

# Revealing the selective mechanisms of inhibitors to PARP-1 and PARP-2 via multiple computational methods

Hongye Hu[1], Buran Chen[2], Danni Zheng[2] and Guanli Huang[1]

[1] Department of Thyroid and Breast Surgery, The First Affiliated Hospital of Wenzhou Medical University, Wenzhou, China
[2] The First Clinical Medical College, Wenzhou Medical University, Wenzhou, China

## ABSTRACT

**Background**. Research has shown that Poly-ADP-ribose polymerases 1 (PARP-1) is a potential therapeutic target in the clinical treatment of breast cancer. An increasing number of studies have focused on the development of highly selective inhibitors that target PARP-1 over PARP-2, its closest isoform, to mitigate potential side effects. However, due to the highly conserved and similar binding sites of PARP-1 and PARP-2, there is a huge challenge for the discovery and design of PARP-1 inhibitors. Recently, it was reported that a potent PARP-1 inhibitor named NMS-P118 exhibited greater selectivity to PARP-1 over PARP-2 compared with a previously reported drug (Niraparib). However, the mechanisms underlying the effect of this inhibitor remains unclear.

**Methods**. In the present study, classical molecular dynamics (MD) simulations and accelerated molecular dynamics (aMD) simulations combined with structural and energetic analysis were used to investigate the structural dynamics and selective mechanisms of PARP-1 and PARP-2 that are bound to NMS-P118 and Niraparib with distinct selectivity.

**Results**. The results from classical MD simulations indicated that the selectivity of inhibitors may be controlled by electrostatic interactions, which were mainly due to the residues of Gln-322, Ser-328, Glu-335, and Tyr-455 in helix αF. The energetic differences were corroborated by the results from aMD simulations.

**Conclusion**. This study provides new insights about how inhibitors specifically bind to PARP-1 over PARP-2, which may help facilitate the design of highly selective PARP-1 inhibitors in the future.

## INTRODUCTION

As one of the most commonly diagnosed malignancies in women, breast cancer accounts for about a quarter of all female cancer cases (*Siegel, Miller & Jemal, 2018*). In the past few years, the incidence of breast cancer has continued to rise, with more than 1 million new cases worldwide each year (*Siegel, Miller & Jemal, 2018*). It is estimated that a total of 5–10% of all breast cancer cases are genetically susceptible to the disease, with multiple breast cancer susceptibility genes having been proposed, including breast cancer 1 (BRCA1) and

Corresponding author
Guanli Huang,
huangguanli@wzhospital.cn

breast cancer 2 (BRCA2), two of the major genes (*Begg et al., 2008*). Currently, sequencing of these two genes is considered as the optimal approach to determining the mutation status in breast cancer patients (*Pfeffer, Ho & Singh, 2017*). Previous studies have shown that homologous recombination-deficient tumor cells resulting from BRCA1 or BRCA2 gene mutations are hypersensitive to the inhibitory effect of poly-ADP ribose polymerase-1 (PARP-1) (*Lin & Kraus, 2017*; *McCann, 2019*). One mechanism that has been proposed as a tentative explanation is that inhibition of PARP-1 blocks DNA single strand break repair and leads to the formation of unrepaired double strand breaks at the replication fork (*Kim et al., 2020*; *Min & Im, 2020*; *Wang, Luo & Wang, 2019b*). Aside from DNA damage repair, PARP-1 is also involved in a wide variety of cellular processes, such as cell proliferation and cell death. The implication of PARP-1 in these processes is a result of the diverse substrates in PARP-1, such as nuclear proteins, which are involved in apoptotic cell death, cell cycle regulation, chromatin decondensation, inflammation, and transcriptional regulation (*Jubin et al., 2016*). Due to these functions, PARP-1 inhibitors have been developed as the first class of cancer therapeutics in clinical trials (*Min & Im, 2020*).

Currently, four PARP-1 drugs have been approved by the US Food and Drug Administration (FDA), including olaparib, niraparib, talazoparib, and rucaparib. In addition to these commercially available PARP-1 drugs, numerous PARP-1 inhibitors have also entered different phases of clinical research, targeting multiple types of tumor either collectively or as single agents (*Min & Im, 2020*). Most of the marketed PARP-1 drugs and inhibitors exhibit poor selectivity when targeting PAPR-1. For instance, olaparib, the first PAPR-1 drug approved by FDA, also interacts with PAPR-1 close homologues PARP-2 and PARP-3 (*Gunderson & Moore, 2015*; *Min & Im, 2020*). Similarly, Niraparib, Talazoparib and rucaparib were also found to exhibit no selectivity between PARP-1 and PARP-2 (*Min & Im, 2020*). There is ample evidence in previous studies that inhibition of PARP-2 could produce potentially undesirable side effects (*Farres et al., 2013*; *Navarro et al., 2017*). For instance, *Farres et al. (2013)* reported that loss of PARP-2 leads to a shortened red blood cell lifespan and impaired differentiation of erythroid progenitor cells, thereby causing chronic anemia. In light of this situation, a great deal of efforts has been put into the design and development of potent PARP-1inhibitors with high selectivity, especially between PARP-1 and PARP-2 (*Eltze et al., 2008*; *Fatima et al., 2014*; *Papeo et al., 2015*). However, given high sequence similarity (84% identity and 90% similarity) and conserved catalytic domain (Figs. 1A–1C), increasing the selectivity of inhibitors remains a huge challenge (*Yelamos et al., 2011*).

Up to now, some potent inhibitors with high selectivity to PARP-1 over PARP-2 have been discovered, such as WD2000-012547, BYK204165 and NMS-P118 (*Eltze et al., 2008*; *Fatima et al., 2014*; *Papeo et al., 2015*). Despite these promising results, little computational research has been conducted to elucidate the selective mechanisms underlying the effect of these inhibitors. To this end, two representative inhibitors (Niraparib, NMS-P118) with divergent selectivity to PARP-1 and PARP-2 were utilized in this study to demonstrate such mechanisms (*Ison et al., 2018*; *Papeo et al., 2015*). Niraparib (also formerly known as MK-4827) is a novel, highly selective, and orally available PARP-1 and PARP-2 small molecule drug developed by Tesaro, approved by FDA in 2017 to treat ovarian cancer

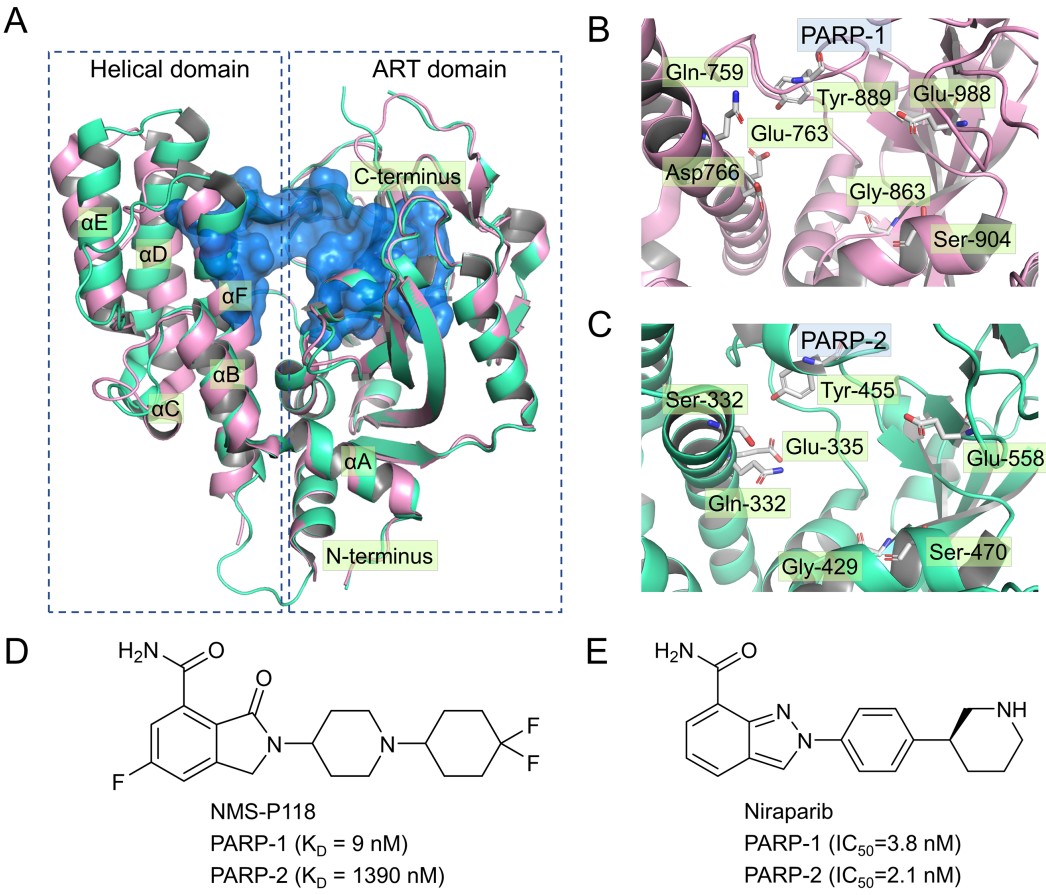

**Figure 1 Alignment of the crystal structures of PARP-1 and PARP-2, and representative inhibitors.**
(A) Overview of the crystal structures of PARP-1 (magenta, PDB code: 5A00) and PARP-2 (green, PDB code: 4ZZY), the binding pocket is colored blue. (B) Close up view of the active site of PARP-1, the residues of Glu-988 is vital for catalysis. (C) Close up view of the active site of PARP-2, the residues of Glu-558 is vital for catalysis. (D) Chemical structure of NMS-P118. (E) Chemical structure of Niraparib.

(*Ison et al., 2018*). Recent clinical studies show that the combined use of niraparib and pembrolizumab can have promising antitumor effects on patients with advanced or metastatic triple-negative breast cancer (*Lin, Yang & Zhao, 2019*). NMS-P118, another inhibitor originally designed by Papeo et al., has been shown to exhibit ~150-fold selectivity to PAPR-1 over PARP2 (Figs. 1D–1E). In addition, the researchers also reported the co-crystal structure of PARP1 and PARP-2 bound to NMS-P118, and proposed that helix αF in the two proteins may be responsible for the drug selectivity, as the induced binding pocket is larger in PARP-1 than in PARP-2 (*Papeo et al., 2015*). However, this explanation seems far from satisfactory. Here, classical molecular dynamics (cMD) simulations and accelerated molecular dynamics (aMD) simulations were employed to clarify the selective mechanisms between PARP-1 and PARP-2 using two representative inhibitors (Niraparib, NMS-P118). A combination of classical MD simulations, the root-mean-square deviations (RMSD), principal component analysis (PCA), dynamical

cross-correlation (DCC) analysis, and root-mean-square fluctuations (RMSF) was applied to investigate the effects of the inhibitors on the protein flexibility and dynamic behavior of key parts of PARP-1 and PARP-2. Afterwards, binding free energy calculations and per-residue free energy decompositions based on the molecular mechanics/generalized Born solvent area (MM/GBSA) method were performed to highlight key residues related to selectivity. Next, aMD simulations combined with RMSD, PCA, DCC and free energy landscape (FEL) analyses were carried out to examine in detail the local energy minima and the conformational space that were not illuminated in the classical MD simulations. Overall, these results can be effective to deepen our understanding of the selective mechanisms between PARP-1 and PARP-2, and may help facilitate the design of novel inhibitors to improve drug selectivity.

## METHODS AND MATERIALS

### Preparation of the initial systems

The three-dimensional structures of human PARP-1 bound to NMS-P118 (PDB ID: 5A00), Niraparib (PDB ID: 4R6E), and PARP-2 bound to NMS-P118 (PDB ID: 4ZZY) were obtained from the Protein Data Bank (*Papeo et al., 2015*; *Thorsell et al., 2017*). The *Loops/Refine Structure* module of *UCSF Chimera* program was employed to model the missing side-chains and loop structures (*Pettersen et al., 2004*). The PDB2PQR Server was employed to estimate the protonation states of ionizable side chains (*Dolinsky et al., 2004*). The initial coordinates of PARP-2 bound to Niraparib were constructed using the AutoDock program (*Morris et al., 2009*). The grid size of cubic box, which was centered on the binding pocket, was set to $60 \times 60 \times 60$ xyz points with a grid spacing of 0.375 Å. The *AutoDockTools* program was employed to assign the Gasteiger partial charges to PARP-2 and Niraparib. The affinity maps of grids were estimated using *AutoGrid* program. The docking protocol involved the generation of 200 conformations. The maximum number of energy evaluations and iterations were set to 25,000,000 and 3,000, respectively. Other parameters were set to default. The top-ranked structure was used for the subsequent molecular dynamics (MD) simulation analyses.

### Classical MD simulation

The prepared crystal structures of PARP-1 bound to NMS-P118 and Niraparib, PARP-2 bound to NMS-P118, and modeled complex of PARP-2 bound to Niraparib were applied to determine the dynamic structural behavior via *Assisted Model Building with Energy Refinement 18* (*Amber 18*) program. The restrained electrostatic potential (RESP) fitting technique was employed to estimate the partial atomic charges of NMS-P118 and Niraparib (*Wang, Cieplak & Kollman, 2000*). The parameters of protein and ligand were derived from the ff14SB force field the General Amber Force Field 2 (GAFF2) in *Amber 18* (*Maier et al., 2015*; *Vassetti, Pagliai & Procacci, 2019*). Each of the prepared complexes was solvated in a cubic box containing TIP3P water molecules, with a minimum distance of 15 Åfrom any edge of the box to any complex atom. Counter ions of an appropriate quantity were added to the system to preserve overall charge neutrality.

Prior to the classical MD simulation, two-step minimizations, heating and equilibration were performed. At first, two-step minimizations were undertaken to eliminate bad contacts between the solvent molecules and the complexes. To reduce the counterions and water molecules to a minimum, a harmonic constraint of 20 kcal mol$^{-1}$ Å$^{-2}$ was first imposed on the four complexes. Then, restriction was eliminated in order for all atoms to move freely. During each stage, the steepest descent minimization of 7,000 steps was performed, followed by conjugate gradient minimization of 7,000 steps. Thereafter, Langevin thermostat with a position restraint of 20 kcal mol$^{-1}$ Å$^{-2}$ was applied to gradually heat up each complex from 0 K to 300 K over 300 ps. Then, each complex was equilibrated at 300 K with 1000 ps simulation time in the isothermal isobaric (NPT) ensemble. Finally, 800 ns production classical MD simulation was carried out for each complex in the NPT ensemble with a time step of 2fs. During the simulations, the Langevin temperature scalings and Berendsen barosta were utilized to maintain the temperature and pressure, respectively (*Izaguirre et al., 2001*; *Praprotnik, Delle Site & Kremer, 2005*). The Particle mesh Ewald (PME) method was employed to estimate the long-range electrostatic interactions, with the cutoff parameter of nonbonded interaction set to 10 Å (*Essmann et al., 1995*). SHAKE method was applied to constrain all covalent bonds connecting hydrogen atoms (*Krautler, VanGunsteren & Hunenberger, 2001*). The coordinates for each complex were saved at an interval of 10 ps for subsequent analysis. The RMSDs and RMSF of the trajectories were calculated using *CPPTRAJ* module in *Amber 18* program.

## aMD simulations

The aMD is an enhanced sampling technique that alters the energy landscape through the addition of a boost potential $\Delta V(r)$ to the original potential energy surface. The $\Delta V(r)$ stands for either the total potential energy ($E_{total}$) or the dihedral energy ($E_{dihedral}$) of a system (*Hamelberg, Mongan & McCammon, 2004*). When $V(r)$ is equal or greater than a predefined reference energy $E$, none of addition energy will be added (Eq. (1)). On the contrary, the modified $\Delta V(r)$ of the system is calculated according to the following Eq. (2):

$$\Delta V(r) = 0 \qquad \Delta V(r) \geq E \tag{1}$$

$$\Delta V(r) = \frac{[E - \Delta V(r)]^2}{\alpha + [E - \Delta V(r)]} \qquad \Delta V(r) < E \tag{2}$$

where $\alpha$ is the acceleration factor that governs the size of the boost. The $\alpha$ is calculated with Eqs. (3) and (4), and the $E$ is calculated according to Eqs. (5) and (6). The boost parameters $E$ and $\alpha$ for the total boost ($E_{total}$ and $\alpha_{total}$) and dihedral boost ($E_{dihedral}$ and $\alpha_{dihedral}$;) are based on the corresponding average $E_{total}$ ($V_{total\_avg}$) and average $E_{dihedral}$ ($V_{dihedral\_avg}$), which are calculated from the classical MD simulations prior to the aMD simulations. $N_{atoms}$ and $N_{res}$ represent the number of atoms and residues in the system, respectively. In Eq. (3), the n is an integer defined as the magnitude of the threshold, which is a multiple of the $\alpha$.

$$E_{total} = V_{total\_avg} + n \times \alpha_{totalral} \tag{3}$$

$$\alpha_{totalral} = 0.2 \times N_{atoms}(N = 1, 2, 3...) \tag{4}$$

$$E_{dihedral} = V_{dihedral\_avg} + (3.5 \times N_{res}) \tag{5}$$

$$\alpha_{dihedral} = 3.5 \times \frac{N_{res}}{5} \tag{6}$$

Herein, the equilibrated structures extracted from classical MD simulations were selected as the initial structures for the aMD simulations. The Dual-boost approach was applied by adding $\Delta V(r)$ to both the $E_{total}$ and $E_{dihedral}$ of the system. The $\Delta V(r)$ was obtained based on the $N_{atoms}$, $N_{res}$, $V_{total\_avg}$, and $V_{dihedral\_avg}$ from the first 40 ns of the classical MD simulations. Then, 800 ns aMD simulations were employed using the dual-boost approach. During aMD simulations, the PME method was employed to estimate the long-range electrostatic interactions, with the cutoff parameter of nonbonded interaction set to 10 Å (*Essmann et al., 1995*). SHAKE method was applied to constrain all covalent bonds connecting hydrogen atoms (*Krautler, VanGunsteren & Hunenberger, 2001*). The Langevin temperature scalings was used to handle the temperature (*Izaguirre et al., 2001*). The coordinates for each complex were saved at an interval of 10 ps and the trajectories were calculated using the *CPPTRAJ* module in *Amber 18* program.

## Principal component analysis (PCA)

PCA was performed on the trajectories from both classical MD and aMD simulations via the *CPPTRAJ* module in *Amber 18* program. All snapshots of each trajectory were aligned to eliminate the translational and rotational motions of all protein $C_\alpha$ atoms. Then, a covariance matrix (3N × 3N) was generated from the Cartesian coordinates. A set of eigenvectors and eigenvalues were generated by diagonalizing the matrix. The top two eigenvalues (principal component 1 and principal component 2, PC1 and PC2) were used for subsequent analysis.

## DCC analysis

The *Bio3D* package of *R* was employed to conduct DCC analysis of the backbone atoms ($C_\alpha$) (*Skjaerven et al., 2014*). The cross-correlation matrix ($C_{ij}$) between residues $i$ and $j$ was generated based on the trajectories from both classical MD simulations and aMD simulations, with 5,000 snapshots for each complex. The $C_{ij}$ is calculated based on the following Eq. (3). The $\Delta r_i$ and $\Delta r_j$ represent the shift from the mean position of the $i$th or $j$th of protein $C_\alpha$ atom. angle bracket represents an average of these two values over the sampled period.

$$C_{ij} = \frac{\langle \Delta r_i \Delta r_j \rangle}{\sqrt{\langle \Delta r_i^2 \Delta r_j^2 \rangle}} \tag{7}$$

## Binding free energy calculations based on classical MD simulations

The MM/GBSA method is often used to perform classical MD simulations combined with free-energy calculations, which can serve as an effective tool for quantitative prediction of protein-ligand binding energies (*Wang et al., 2019a*). The binding free energies ($\Delta G_{\text{bind}}$) are calculated on the basis of the MM/GBSA method according to the following equations:

$$\Delta G_{bind} = \Delta G_{R+L} - (\Delta G_R + \Delta G_L) = \Delta E_{MM} + \Delta G_{sol} - T\Delta S \tag{8}$$

$$\Delta E_{MM} = \Delta E_{int} + \Delta E_{vdW} + \Delta E_{elec} \tag{9}$$

$$\Delta G_{sol} = \Delta G_{GB} + \Delta G_{SA} \tag{10}$$

In Eq. (8), $\Delta G_{R+L}$, $\Delta G_R$, and $\Delta G_L$ stand for the free energies of receptor–ligand complex, receptor and ligand, respectively. The sum of molecular mechanics interaction energy ($\Delta E_{MM}$), solvation energy ($\Delta G_{sol}$) and the change of the conformational entropy at temperature T ($-T\Delta S$) are equal to the sum of $\Delta G_{R+L}$, $\Delta G_R$, and $\Delta G_L$. In Eq. (9), $\Delta E_{MM}$ is given as the sum of intermolecular interaction energy ($\Delta E_{int}$), van der Waals energy ($\Delta E_{vdW}$), and electrostatic energy ($\Delta E_{elec}$). The solvation free energy can be divided into polar ($\Delta G_{GB}$) and nonpolar ($\Delta G_{SA}$) and contributions (Eq. (10)). Here, for each complex, 1,000 structures extracted from the classical MD simulations with a simulation time between 600 and 800 ns were utilized to conduct binding free energy calculations and per-residue energy decomposition. The $\Delta E_{int}$ was canceled as the single trajectory strategy was executed. Based on parameters by Onufriev et al., the $\Delta G_{GB}$ was determined using a modified GB model (GB$^{OBC1}$) (*Onufriev, Bashford & David, 2000*). The $\Delta G_{SA}$ was determined using the solvent accessible surface area (SASA) model ($\Delta G_{SA} = \sigma * \text{SASA}$). The parameter $\sigma$ was set to 0.0072 kcal mol$^{-1}$ Å$^{-2}$. The $-T\Delta S$ was excluded from consideration because of relatively low prediction accuracy and high computational demand (*Hou et al., 2011*).

## Free energy landscape (FEL) calculation based on aMD

The cumulant expansion to the second order method was employed to determine the FEL for each simulated complex from aMD simulations, because this method offers a good approximation for calculating the reweighting factor. In this study, the $\Delta V(r)$ combined with PC1 and PC2 from PCA of the aMD simulation trajectories were utilized to recover the FEL (*Miao et al., 2014b*; *Roe & Cheatham 3rd, 2013*).

## RESULTS

### Evaluation of the stability of simulated complexes from classical MD simulations

To validate the docking results of the modeled complex of PARP-2 bound to Niraparib, structural alignments of the PARP-2/NMS-P118 and modeled PARP-2/Niraparib were

A

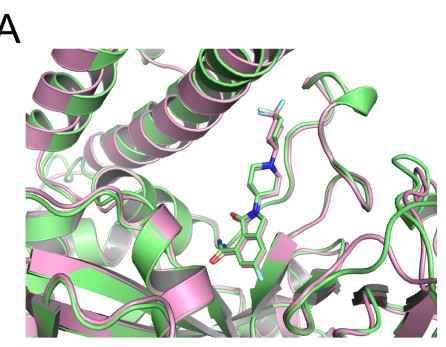

PARP-1/NMS-P118 (magenta)
PARP-2/ NMS-P118 (green)

B

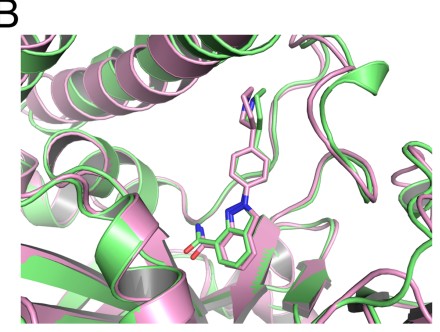

PARP-1/Niraparib (magenta)
PARP-2/Niraparib (green)

**Figure 2** **Validation of modeling results.** (A) Alignment of the crystal structures of PARP-1/NMS-P118 and PARP-2/NMS-P118. (B) Alignment of the crystal structure of PARP-1/Niraparib and the modeled structure of PARP-2/Niraparib.

performed with the corresponding crystal structures of PARP-1/NMS-P118 and PARP-1/Niraparib. As shown in Fig. 2A, alignment of the crystal structures between PARP-1/NMS-P118 (PDB code: 5A00) and PARP-2/NMS-P118 (PDB code: 4ZZY) exhibited relatively high similarity, with a RMSD of 0.704 Åfor heavy atoms. Similarly, the alignment of the modeled complex of PARP-2/Niraparib with the crystal structure of PARP-1/Niraparib (PDB code: 4R6E) mostly showed similarities with only minor differences, with a RMSD of 0.767 Å for heavy atoms (Fig. 2B). These results suggest that the predicted model is sufficient to study the dynamic features through further MD simulations.

Firstly, 800 ns classical MD simulations for the modeled structure and the three crystal structures were performed. As a prerequisite for all further analyses, the dynamic stability of the simulated complexes was monitored by studying the RMSDs for all protein backbones ($C_\alpha$) atoms and all ligand heavy atoms for each complex with the starting structure as a function of simulation time. Theoretically, smaller fluctuations of RMSDs indicate greater stability of the complex. As shown in Fig. 3, the time evolution of the RMSD values of $C_\alpha$ atoms and ligand heavy atoms in each complex tend to converge after ~100-300 ns simulations. During the simulation of the last 400 ns, the RMSD curves of both PARP-1 $C_\alpha$ atoms displayed minor fluctuations (<1 Å), indicating that NMS-P118 constrained the protein structural flexibility of PARP-1. In contrast, those for both PARP-2 and NMS-P118 exhibited greater fluctuations compared to the complex of PARP-2/NMS-P118 (Fig. 3B). This attested to the highly unstable nature of PARP-2 when it was bound to selective PARP-1 inhibitor NMS-P118. In comparison, the RMSD curves for Niraparib in both PARP-1 and PARP-2 oscillated with minute fluctuations (<1 Å) during the last 400 ns simulation, indicating that the non-selective drug Niraparib constrained the structural flexibility of both PARP-1 and PARP-2. Based on these results, the structural and energetic properties for each complex were further analyzed in detail.

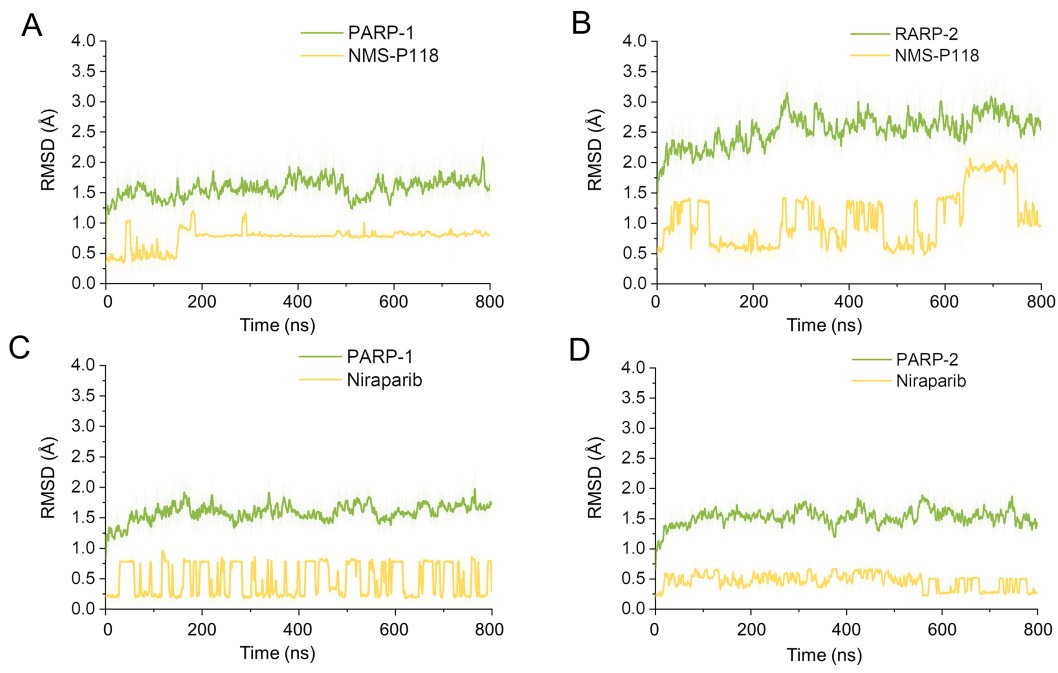

**Figure 3** **RMSD values of the protein backbones atoms and ligand heavy atoms from the 800 ns classical MD simulations.** (A) PARP-1/NMS-P118. (B) PARP-2/NMS-P118. (C) PARP-1/Niraparib. (D) PARP-2/Niraparib.

## Dynamic features of each complex from classical MD simulations

The different flexibility for NMS-P118 and Niraparib in different proteins may result in different protein conformational transitions. To characterize the protein conformational transitions over time, PCA was performed to identify the trend in large-scale collective motions. Generally, the eigenvectors with the highest eigenvalues capture the majority of variance in the original protein conformational distributions (*Mashiko, 2018*; *Sittel, Filk & Stock, 2017*). Herein, the protein conformational ensembles were investigated by projecting the first two principal components (PC1 and PC2) from classical MD simulations onto a two-dimensional space. Generally, when PC1 and PC2 are plotted against each other, structures with high similarity are clustered together. As a result, a cluster represents a different state of protein conformation. As shown in Fig. 4, the protein conformational distributions for each complex were dynamic during 800 ns classical MD simulations and eventually reached overall stability. It is apparent that the conformational distributions of PARP-1/NMS-P118 were remarkably different from those of PARP-2/NMS-P118, while those for PARP-1/Niraparib and PARP-2/Niraparib were also different. Analysis of the protein conformational distributions clearly showed that PARP-2/NMS-P118 samples a wider conformational space compared to PARP-1/NMS-P118. However, those for PARP-1/Niraparib and PARP-2/Niraparib shared a certain degree of similarity. These results demonstrate that the selective PARP-1 inhibitor NMS-P118 bound to PARP-1 had a different protein conformational flexibility compared to PARP-1, not the non-selective drug Niraparib.

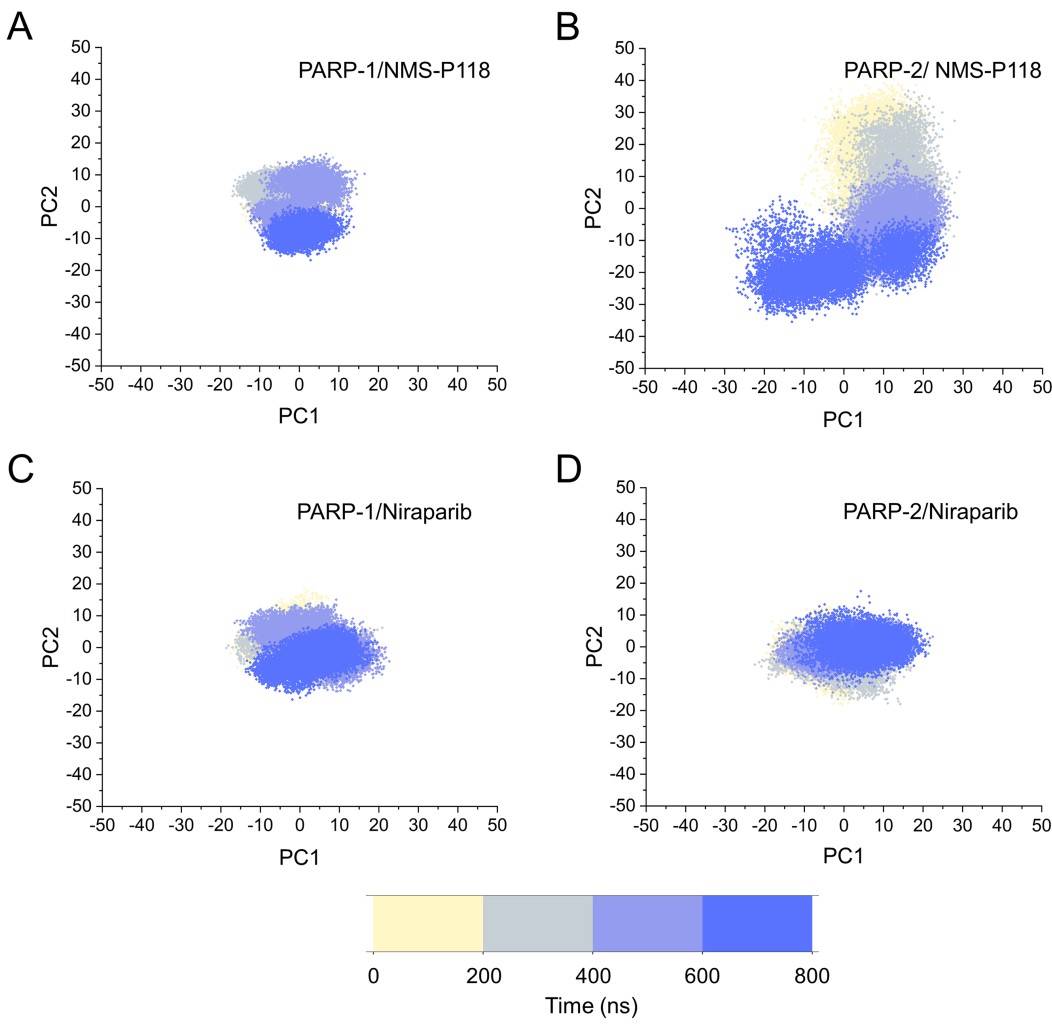

**Figure 4** **The top two ranked principal components (PC1, PC2) are plotted against each other from the 800 ns classical MD simulations.** (A) PARP-1/NMS-P118. (B) PARP-2/NMS-P118. (C) PARP-1/Niraparib. (D) PARP-2/Niraparib.

To further investigate the correlation between the motion of the residues in the proteins, DCC analysis was performed. As plotted in Fig. 5, the direction of correlation is represented by a color gradient ranging from blue (negative correlation) to red (positive correlation). The correlation coefficient ($-1$ to $+1$), corresponding to three different colors: dark blue ($-0.25$ to $-1$) represents anti-correlation; dark red (0.25 to 1) represents positive correlation; blue represents anti-correlation ($-0.25$ to $-1$); and light red or light blue ($-0.25$ to $+0.25$) represents weak or no-correlation. It can be observed that both the red and blue regions in PARP-2/NMS-P118 were larger and more intense than those in PARP-1/NMS-P118, implying elevated correlation or anti-correlation motions in PARP-2/NMS-P118 (Figs. 5A and 5B). In comparison, color regions for PARP-1 and PARP-2 bound to the non-selective drug Niraparib were quite similar (Figs. 5C and 5D). These

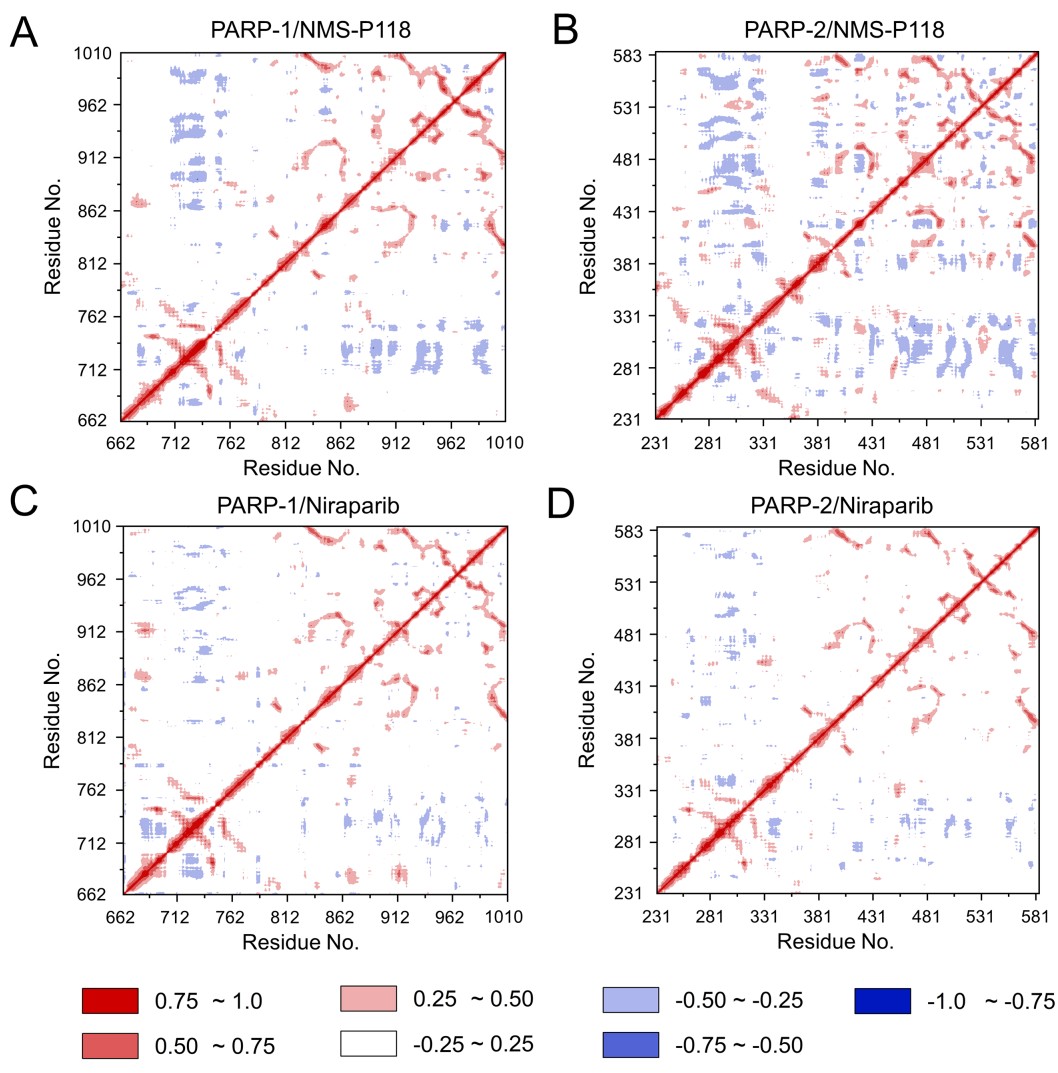

**Figure 5   DCC analysis from the 800 ns classical MD simulations.** (A) PARP-1/NMS-P118. (B) PARP-2/NMS-P118. (C) PARP-1/Niraparib. (D) PARP-2/Niraparib.

results indicate that the relative motions of different protein subdomains may correlate with different protein flexibility, which was responsible for drug selectivity.

To further highlight the key sub-domains, the $C_{\alpha}$ of RMSF analyses were conducted. Low RMSF values of residues represented less flexibility, whereas high RMSF values indicated greater fluctuations in relation to their average position during simulation. As shown in Fig. 6, the RMSFs of PARP-1/NMS-P118 showed a high degree of similarity with those of PARP-1/Niraparib. However, the helix αF connected with loop 347–356 in PARP-2 exhibited amplified fluctuations when it binds to the selective PARP-1 inhibitor NMS-P118 compared with the non-selective drug Niraparib. The above results indicate that the binding of selective PARP-1 inhibitor NMS-P118 leads to increased fluctuations

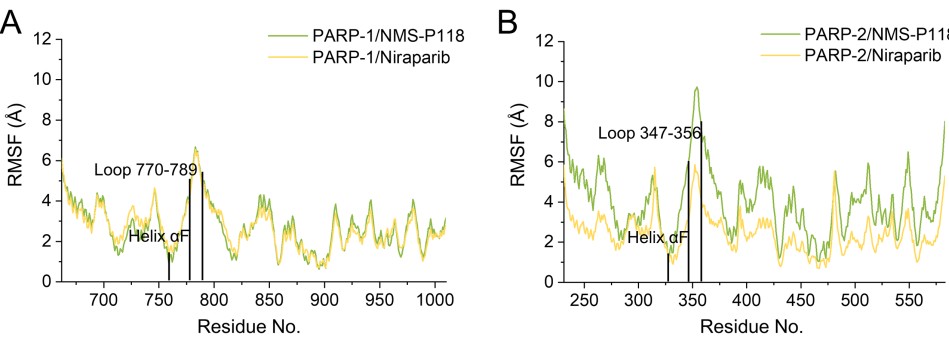

**Figure 6** **RMSF analysis from the 800 ns classical MD simulations.** (A) Alignment of RMSF of PARP-1/NMS-P118 and PARP-1/Niraparib. (B) Alignment of RMSF of PARP-2/NMS-P118 and PARP-2/Niraparib.

of PARP-2 (especially the αF and loop 347–356), which might be the dominating driving force for the redistributions of free energies.

## Binding free energy calculations using the MM/GBSA method based on classical MD simulation trajectories

MM/GBSA binding free energy calculations based on classical MD simulation trajectories were applied for the assessment of the energy properties of NMS-P118 and Niraparib when they were bound with PARP-1 or PARP-2. As listed in Table 1, the predicted binding free energies ($\Delta G_{\text{binding}}$) for PARP-1/NMS-P118, PARP-2/NMS-P118, PARP-1/Niraparib and PARP-2/Niraparib were −46.06 ± 0.15, −36.40 ± 0.14, −44.35 ± 0.22, and −43.89 ± 0.24 kcal/mol, respectively. It is clear that the $\Delta G_{\text{binding}}$ was highly correlated with the experimental data reported, and that for each system, different energy terms contribute differentially to the $\Delta G_{\text{binding}}$. In this study, only the polar contributions ($\Delta E_{\text{elec}} + \Delta G_{\text{GB}}$) for the most vital factor were discussed. The polar contributions for the PARP-1/NMS-P118 and PARP-2/NMS-P118 were significantly different at 10.65 ± 0.34 and 19.29 ± 0.16 kcal/mol, respectively. In comparison, those for PARP-1/Niraparib and PARP-2/Niraparib were quite similar at 9.72 ± 0.29 and 9.84 ± 0.17 kcal/mol. The non-polar contributions ($\Delta E_{\text{vdW}} + \Delta G_{\text{SA}}$) for PARP-1/NMS-P118 and PARP-2/NMS-P118 (−56.70 ± 0.12 and −55.69 ± 0.19 kcal/mol) were almost identical with those for PARP-1/Niraparib and PARP-2/Niraparib (−53.92 ± 0.26 and −53.72 ± 0.41 kcal/mol). Taken together, these results demonstrate that the polar contribution has a significant impact on drug selectivity to PARP-1 and PARP-2.

To gain further insights into the vital residues in drug selectivity, per-residue decomposition based on the MM/GBSA method was employed to assess residue contributions to the binding of the protein-ligand complexes. Per-residue energy differences between Niraparib and NMS-P118 systems ($\Delta\Delta G = \Delta G_{\text{Niraparib}} - \Delta G_{\text{NMS-P118}}$) were plotted to identify the key residues. Negative values represent the residues of Niraparib formed stronger interactions with the protein than those of NMS-P118, whereas positive values indicated quite the opposite, namely, that the residues of Niraparib formed weaker interactions with the protein than those of NMS-P118. As shown in Fig. 7A, the differences

**Table 1   Binding free energies of NMS-P118 and Niraparib in PARP-1 and PARP-2 (kcal/mol).**

| Ligand | NMS-P118 | | Niraparib | |
| --- | --- | --- | --- | --- |
| Protein | PARP-1 | PARP-2 | PARP-1 | PARP-2 |
| $\Delta E_{vdW}$ | $-50.94 \pm 0.14$ | $-49.87 \pm 0.12$ | $-48.65 \pm 0.16$ | $-48.46 \pm 0.14$ |
| $\Delta E_{elec}$ | $-85.06 \pm 0.70$ | $-54.88 \pm 0.53$ | $-116.24 \pm 0.91$ | $-86.98 \pm 1.05$ |
| $\Delta G_{GB}$ | $95.71 \pm 0.68$ | $74.17 \pm 0.52$ | $125.96 \pm 0.78$ | $96.82 \pm 0.89$ |
| $\Delta G_{SA}$ | $-5.76 \pm 0.01$ | $-5.82 \pm 0.01$ | $-5.27 \pm 0.01$ | $-5.26 \pm 0.01$ |
| $\Delta E_{nonpolar}$ | $-56.70 \pm 0.12$ | $-55.69 \pm 0.19$ | $-53.92 \pm 0.26$ | $-53.72 \pm 0.41$ |
| $\Delta E_{polar}$ | $10.65 \pm 0.34$ | $19.29 \pm 0.16$ | $9.72 \pm 0.29$ | $9.84 \pm 0.17$ |
| $\Delta G_{bind}$ | $-46.06 \pm 0.15$ | $-36.40 \pm 0.14$ | $-44.35 \pm 0.22$ | $-43.89 \pm 0.24$ |

**Notes.**

$\Delta E_{vdW}$, Van der Waals energy; $\Delta E_{ele}$, Electrostatic energy; $\Delta G_{GB}$, Electrostatic contribution to solvation; $\Delta G_{SA}$, Non-polar contribution to solvation; $\Delta E_{nonpolar}$, Non-polar interaction; $\Delta E_{polar}$, polar interaction; $\Delta G_{bind}$, Binding free energy.

of $\Delta\Delta G$ between PARP-1/NMS-P118 and PARP-1/Niraparib were quite small with $\Delta\Delta G$ less than 0.5 kcal/mol. Alignment of the representative structures of PARP-1/NMS-P118 and PARP-1/Niraparib were highly similar (Fig. 7B). However, the residues of Ser-328, Gln-322, Glu-335, and Tyr-455 formed significantly stronger interactions with Niraparib than with NMS-P118 in PARP-2 (Fig. 7C). Notably, the key residues of Ser-328, Gln-322 and Glu-335 were located in the helix αF, which exhibited amplified fluctuations in RMSF analysis. This may be due to the fact that the distinctive flexibility of the helix αF in PARP-2 bound to selective PARP-1 inhibitors induced energetic redistributions.

## Evaluation of the stability of the simulated complexes from aMD simulations

To explore the conformational behaviors in greater detail, aMD simulations were first employed. Thereafter, the RMSDs of $C_\alpha$ atoms and ligand heavy atoms were monitored. As shown in Fig. 8, the RMSD values of $C_\alpha$ atoms and ligand heavy atoms for each complex reached equilibrium after 120-200 ns aMD simulations, suggesting that simulated complexes became dynamically stable through 800 ns aMD simulations. Interestingly, the fluctuations of selective PARP-1 inhibitor NMS-P118 bound to PARP-1 were much smaller than when bound to PARP-2. In contrast, the fluctuations of non-selective drug Niraparib were similar in both PARP-1 and PARP-2. A comparison of these results revealed that the selective PARP-1 inhibitor allowed for larger protein conformational changes of PARP-2. These findings were corroborated by the DCC analysis, which further revealed that the red and blue regions in PARP-2/NMS-P118 were both larger and more intense than those in PARP-1/NMS-P118, while those for PARP-1 and PARP-2 bound to Niraparib were quite similar (Fig. 9).

Based on the above findings, PCA was used to identify the various protein conformations obtained during the aMD simulations. As shown in Fig. 10, the protein conformations of for each complex was characterized by dynamic fluctuations during 800 ns aMD simulations. Similar with the results from classical MD simulaitons, the conformational

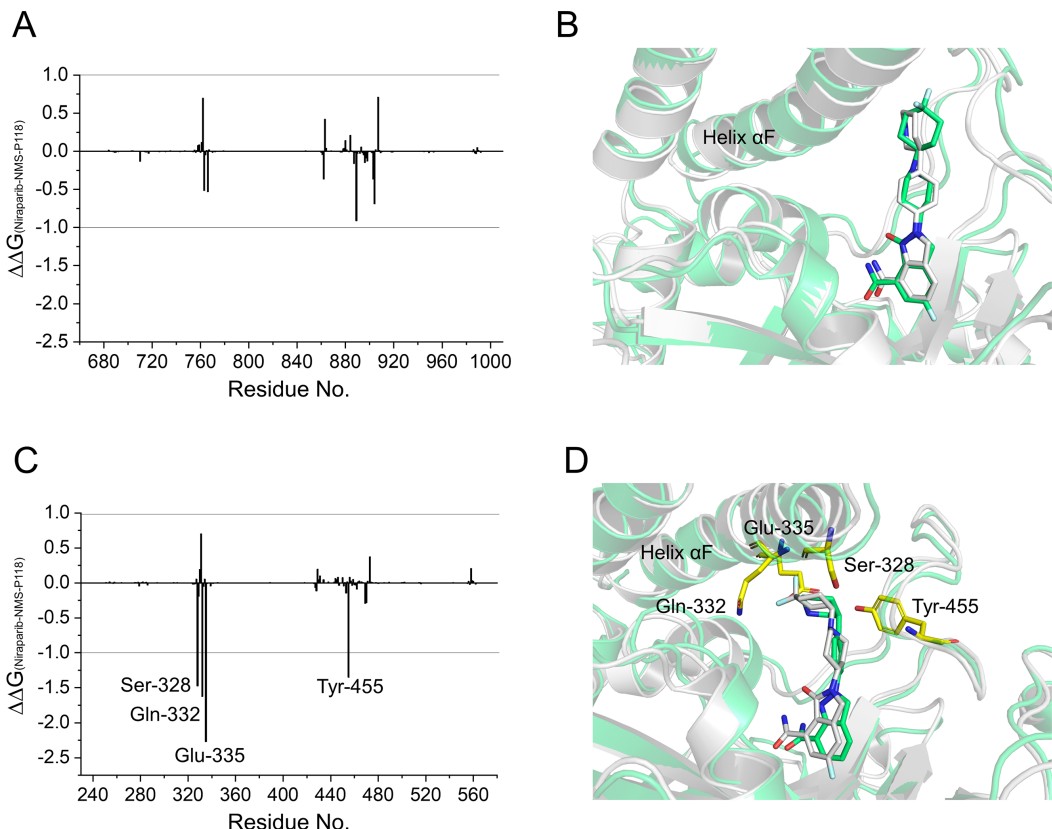

**Figure 7 Energy difference in per-residue contributions between Niraparib and NMS-P118.** (A) The energy differences between PARP-1/NMS-P118 (green) and PARP-1/Niraparib (gray). (B) Alignment of representative structures of PARP-1/NMS-P118 and PARP-1/Niraparib. (C) The energy differences between PARP-2/NMS-P118 and PARP-2/Niraparib. (D) Alignment of representative structures of PARP-2/NMS-P118 (green) and PARP-2/Niraparib (gray).

distributions of PARP-1/NMS-P118 were remarkably different from those of PARP-2/NMS-P118 (Figs. 10A–10B). Meanwhile, those for PARP-1/Niraparib and PARP-2/Niraparib were also different (Figs. 10C–10D). Compared to the PARP-1/NMS-P118 complex, the PARP-2/NMS-P118 complex exhibited more structural clusters and a wider range of conformational distributions (Figs. 10A–10B). However, the conformational distributions for Niraparib bound to PARP-1 and PARP-2 had a somewhat similar range (Figs. 10C–10D), suggesting that the selective PARP-1 inhibitors could stabilize the protein conformation of PARP-1. These results were in agreement with the RMSDs, DCC and PCA analyses from the classical MD simulations.

The FEL was employed to further demonstrate the relationship between the changes of conformation and energy (Fig. 11). Generally, more energy wells (dark blue regions) represent greater conformational changes of the protein during aMD simulation (*Han et al., 2019*; *Miao, Nichols & McCammon, 2014a*). As shown in Fig. 11A, only a deep energy well for NMS-P118 bound to PARP-1 was observed throughout the whole 800 ns aMD simulation. In contrast, two major deep energy wells with a much wider

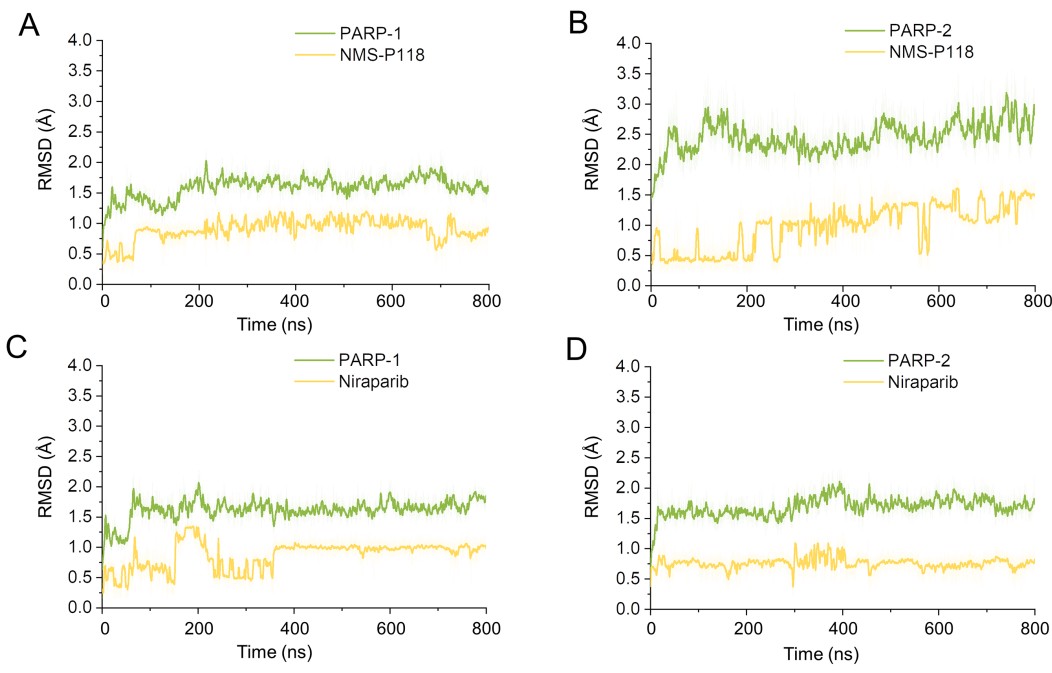

**Figure 8** **RMSD values of the protein backbones atoms and ligand heavy atoms from the 800 ns aMD simulations.** (A) PARP-1/NMS-P118. (B) PARP-2/NMS-P118. (C) PARP-1/Niraparib. (D) PARP-2/Niraparib.

range of distributions were observed for the complex of PARP-1/NMS-P118 (Fig. 11B). Nevertheless, those for Niraparib bound to PARP-1 and PARP-2 showed a similar range of distributions and both were confined to a single deep energy. These results highlight the unstable nature of PARP-2 bound to selective PARP-1 inhibitor NMS-P118.

## DISCUSSION

PARP-1 inhibitors have been widely studied as potential cancer therapeutics for breast and ovarian cancers (*Min & Im, 2020*). The reported clinical candidates and preclinical PARP-1 inhibitors were designed with the purpose of imitating the nicotinamide portion of nicotinamide adenine dinucleotide (NAD$^+$), with which they compete for the corresponding PARP-1 binding site. A number of recent studies have indicated that due to their high sequence and structural similarity (Figs. 1A–1C), non-selective PARPs drugs may result in potentially undesirable side effects, especially between PARP-1 and PARP-2 (*Eltze et al., 2008*; *Fatima et al., 2014*; *Papeo et al., 2015*). In this aspect, the ideal drug candidate should be a highly selective PARP-1 inhibitor with greater subtype specificity. Until now, only a few highly selective PARP-1 inhibitors have been reported. However, even fewer computational modeling researches have been conducted to elucidate their underlying selective mechanisms.

In the present study, a comprehensive molecular computational method was employed to demonstrate the selective mechanisms via two representative inhibitors (NMS-P118,

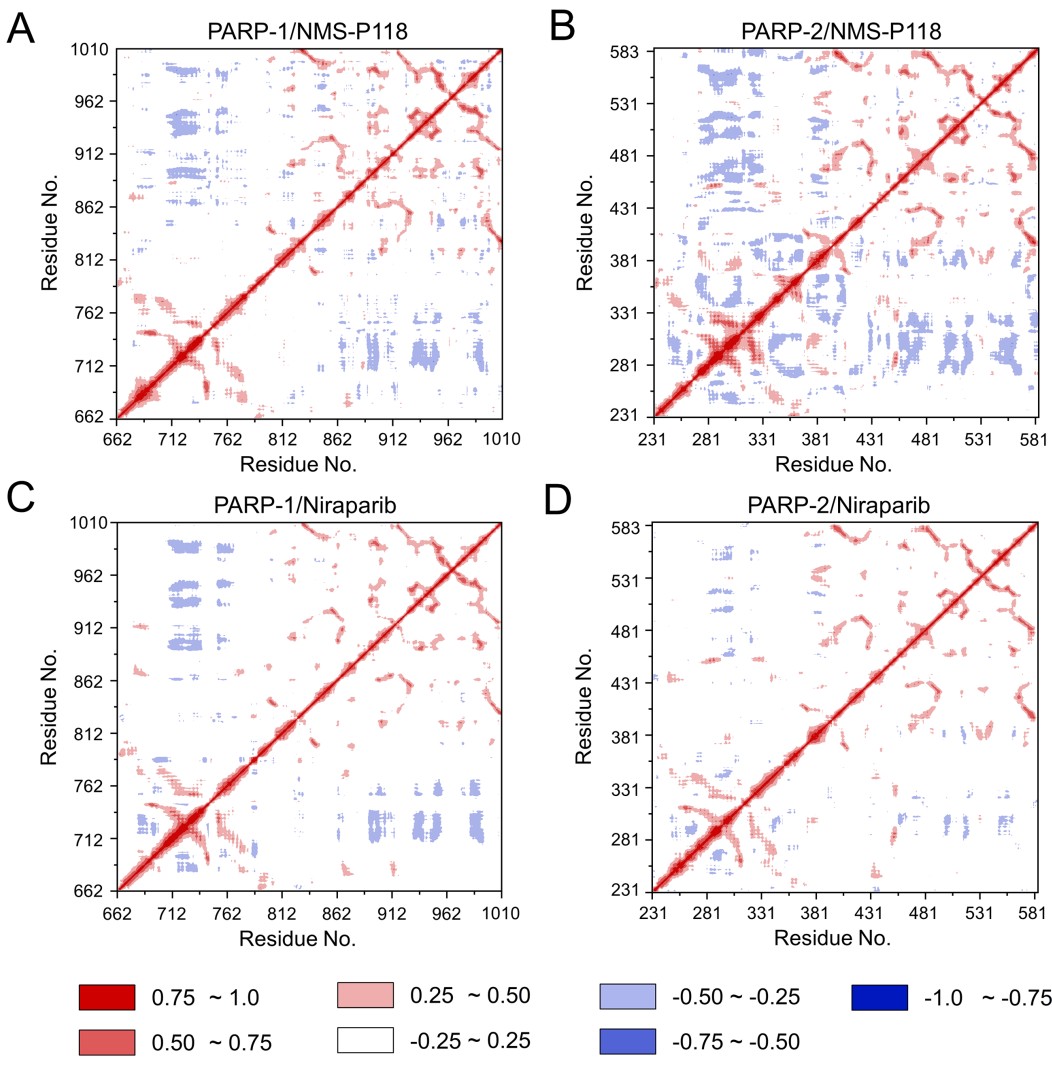

**Figure 9** **DCC analysis from the 800 ns aMD simulations.** (A) PARP-1/NMS-P118. (B) PARP-2/NMS-P118. (C) PARP-1/Niraparib. (D) PARP-2/Niraparib.

Niraparib) with different selectivity to PARP-1 and PARP-2 (Figs. 1D–1E). Initially, molecular docking was applied to predict the complex of PARP-2/Niraparib. To examine the prediction accuracy, the predicted and the crystal structures were aligned. The RMSDs between the predict pose (PARP-2/Niraparib) and the crystal structure (PARP-1/Niraparib) were highly similar, a finding that was consistent with the results observed by alignment of crystal structures of NMS-P118 bound to PARP-1 and PARP-2 (Fig. 2). Based on the above findings, these structures were employed to further explore the dynamic behavior via classical MD simulations and aMD simulations. Classical MD simulations in conjunction with the RMSD, PCA, and DCC analyses provided compelling evidence that the conformation fluctuations were different for PARP-2 bound to selective PARP-1 inhibitors and PARP-2 bound to non-selective inhibitor (Figs. 3–5). Further RMSF analysis

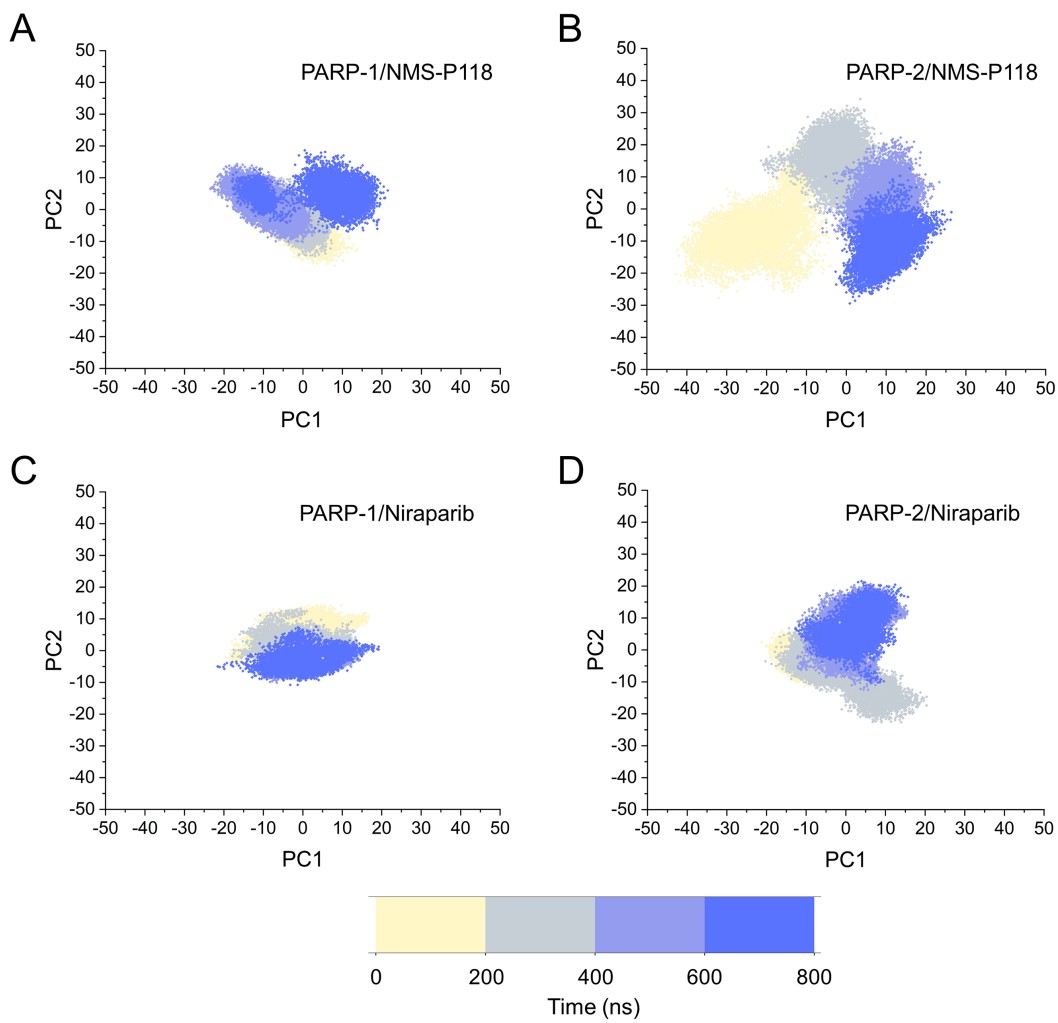

**Figure 10** **The top two ranked principal components (PC1, PC2) are plotted against each other from the 800 ns aMD simulations.** (A) PARP-1/NMS-P118. (B) PARP-2/NMS-P118. (C) PARP-1/Niraparib. (D) PARP-2/Niraparib.

revealed that the major structural variations were the conformational changes of the helix αF, which may account for drug selectivity (Fig. 6). According to the binding free energy calculations, the polar contributions had an obvious impact on drug selectivity (Table 1). Per-residue decomposition analysis further revealed that drug selectivity was primarily controlled by the residues of Ser-328, Gln-322, Glu-335, and Tyr-455, most of which are located in the helix αF (Fig. 7).

As there are possible energy barriers between various meta-stable states, the classical MD simulations may still not be sufficient to sample the possible conformations (*Hamelberg, Mongan & McCammon, 2004*; *Miao, Nichols & McCammon, 2014a*). Therefore, an enhanced sampling technique that can sample the protein conformation at various meta-stable states is still required. Most of enhanced sampling techniques often require expert knowledge of the studied complexes, as these techniques require reaction coordinates. This

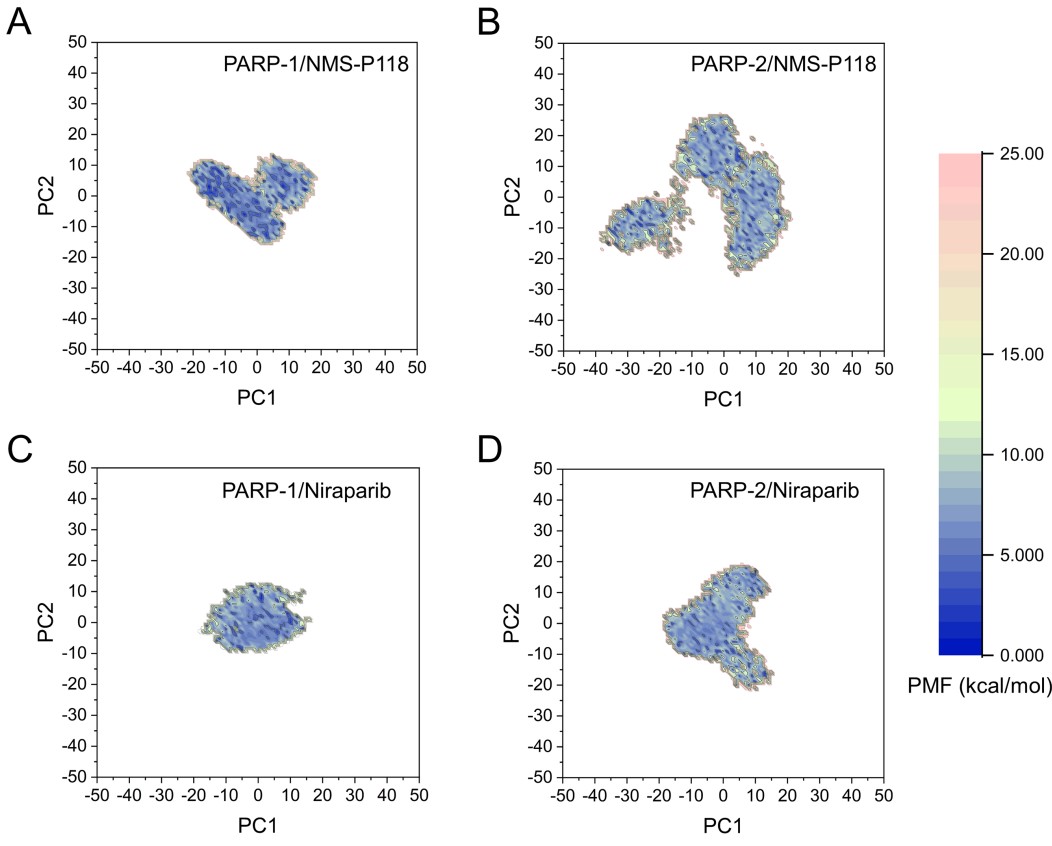

**Figure 11 FEL analysis from the 800 ns aMD simulations.** (A) PARP-1/NMS-P118. (B) PARP-2/NMS-P118. (C) PARP-1/Niraparib. (D) PARP-2/Niraparib.

limitation is overcome by the aMD simulations, which can explore the conformational behavior of the biomacromolecule without this requirement (*Hamelberg, Mongan & McCammon, 2004*; *Miao, Nichols & McCammon, 2014a*). Therefore, within the framework of this study, aMD simulation was employed to further sample the possible conformational ensembles. The results of RMSD, PCA, DCC analyses from aMD simulations indicated that the selective PAPR-1 inhibitor NMS-P118 may significantly disrupt the stability of the PARP-2 protein, not the non-selective drug Niraparib, a finding that was consistent with the results from classical MD simulations (Figs. 8–10). The FEL analysis further demonstrated the unstable nature of PAPR-2 bound to the selective PAPR-1 inhibitors NMS-P118 (Fig. 11). The preferential binding of NMS-P118 to PARP-1 takes precedence over PARP-2, resulting in a better inhibitory effect of PARP-1 than PARP-2, and greater PARP-1 selectivity. In summary, more selective PAPR-1 inhibitors may be required to evaluate the above findings via molecular modeling, which might help facilitate the rational design of high selective PAPR-1 inhibitors.

## CONCLUSIONS

Multiple computational techniques were employed to conduct an exploration of the selective mechanisms of inhibitors underlying PARP-1 and PARP-2. The results from classical MD simulations offered compelling evidence that preferential NMS-P118 binding to PARP-1 over PARP-2 was controlled by the protein conformational changes of helix αF, which lead to decreased polar contributions. These findings were further corroborated by the RMSDs, DCC, PCA analyses from aMD simulations. FEL results from aMD simulations further suggested that PAPR-2 bound to the selective PARP-1 inhibitor NMS-P118, but not the non-selective drug Niraparib, underwent large conformational changes. Taken together, these results may prove conducive to the design of more selective PAPR-1 inhibitors with fewer potential side effects.

### Funding

This work was supported by the Natural Science Foundation of Zhejiang Province of China (LY20H160011). The funders had no role in study design, data collection and analysis, decision to publish, or preparation of the manuscript.

### Grant Disclosures

The following grant information was disclosed by the authors:
Natural Science Foundation of Zhejiang Province of China: LY20H160011.

### Competing Interests

The authors declare there are no competing interests.

### Author Contributions

- Hongye Hu conceived and designed the experiments, performed the experiments, prepared figures and/or tables, and approved the final draft.
- Buran Chen conceived and designed the experiments, performed the experiments, analyzed the data, authored or reviewed drafts of the paper, and approved the final draft.
- Danni Zheng analyzed the data, authored or reviewed drafts of the paper, and approved the final draft.
- Guanli Huang conceived and designed the experiments, analyzed the data, prepared figures and/or tables, authored or reviewed drafts of the paper, and approved the final draft.

### Data Availability

  All raw data are available in the Supplemental Files.

### Supplemental Information

Supplemental information for this article can be found online at http://dx.doi.org/10.7717/peerj.9241#supplemental-information.

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
