# Peer review of "Revealing the selective mechanisms of inhibitors to PARP-1 and PARP-2 via multiple computational methods"

_PeerJ, doi:10.7717/peerj.9241_

## Round 0.1 · original submission · Minor Revisions

Our reviewers have only minor requests for you. I look forward to receiving a final version!

Reviewer 1 ·

Basic reporting

In the manuscript titled “Revealing the selective mechanisms of inhibitors to PARP-1 and PARP-2 via multiple computational methods”. The authors performed a series of molecular modeling techniques to reveal the selective mechanisms. The article deals with a very interesting topic with quite understandable and comprehensively explained.

Experimental design

The computational methods of docking, classical MD simulations, aMD simulations follow well-established protocols, and are appropriate for explore the selective mechanisms of inhibitors to PARP-1 and PARP-2.Some minor revisions are still requested before it can be published.

1. The format of △EvdW, △Eelec in table 1 is inconsistent with in notes, please change the capital English letters to italic.
2. The number of equations in line 210-217 is not correct. As an example, the equation in line 210 should be “(7)”.

Validity of the findings

The raw data were provided and the results were reliable.

Additional comments

Overall this paper meets the criteria for publication with minor revisions.

Reviewer 2 ·

Basic reporting

Pass: the manuscript is generally well-written, and the data presented in a clear and readily-understood fashion by experts in the field.

Experimental design

Pass: The research question (explaining the molecular-level mechanism of selective inhibition) is well defined and meaningful. The methods employed are well-established and appears rigorously applied.

Validity of the findings

Pass: The conclusions reached are consistent with the results obtained from simulations, and are valid insofar as the basic assumptions underlying the simulation methodology itself (such as forcefield employed, timescale sampled) are valid.

Additional comments

This manuscript details a set of computational approaches employed to explore the selective
inhibition mechanisms of the ligand NMS-P118, and attempts to provide molecular-level explanations for its selective inhibition of PARP-1 over PARP-2. Classical equilibrium MD simulations suggest that preferential NMS-P118 binding to PARP-1 over PARP-2 was controlled by the conformational changes of helix alpha-F. Accelerated MD simulations, supplemented by free energy calculations of the conformations of the proteins expressed in principal component space (PC1 vs PC2), provide further details of the conformational changes induced by binding of PARP-1 and PARP-2 to the ligands NMS-P118 and Niraparib.

The methodology adopted is well-established and appears rigorous, with sufficient simulation trajectory lengths employed. The manuscript is generally well-written and the data presented clearly. I recommend publication in its present form. There is a minor error: the x-axis of Figure 10A should read "PC1".

Reviewer 3 ·

Basic reporting

The work is presented well

Experimental design

The manuscript presents a comprehensive computational studies of the complexes formed by PARP-1 (an important drug target) and PARP-2 with selective and nonelective inhibitors. The work is relevant and rigorous.

Validity of the findings

no comment

Additional comments

In this manuscript Hu et al. present a comprehensive computational study on the mechanisms underlaying selective inhibition of PARP-1 over PARP-2 from the small-molecule inhibitor NMS-P118. I think the overall study is well performed and deserves to be published in PeerJ after the following minor issues are considered:


1) The authors should discuss better their findings in the context of the inhibitory mechanisms. What is the inhibitory mechanism? Competitive? Non-competitive? Other? How does the higher PARP-2 flexibility induced by NMS-P118 affect inhibition?
2) Figure 1A should contain a close up view of the active site with displayed residues that are important for catalysis
3) The authors mentioned the existence of a few non-specific (Niraparib, Talazoparib and rucaparib) and specific (WD2000-012547, BYK204165 and NMS-P118) PARP-1 inhibitors? Why they only focused on Niraparib and NMS-P118?

---

## Round 0.2 · accepted · Accept

I am glad to accept your manuscript. I look forward to seeing it in print!

Reviewer 1 ·

Basic reporting

no comment

Experimental design

no comment

Validity of the findings

no comment

Additional comments

no comment

Reviewer 3 ·

Basic reporting

The authors have addressed my concerns

Experimental design

The authors have addressed my concerns

Validity of the findings

The authors have addressed my concerns

Additional comments

The authors have addressed my concerns